# Analytical and Numerical Modeling of the Pullout Behavior between High-Strength Stainless Steel Wire Mesh and ECC

**DOI:** 10.3390/ma15165649

**Published:** 2022-08-17

**Authors:** Xuyan Zou, Yawen Liu, Juntao Zhu, Ke Li, Jinglong Cao

**Affiliations:** 1Department of Civil Engineering, Zhengzhou Institute of Technology, Zhengzhou 450044, China; 2Department of Civil Engineering, Zhengzhou University, Zhengzhou 450001, China

**Keywords:** bond–slip relationship, engineering cementitious composites, finite element model, high-strength stainless steel wire mesh

## Abstract

Bond behavior is a key factor in the engineering application of composite material. This study focuses on the constitutive model of the bond behavior between high-strength stainless steel strand mesh and Engineered Cementitious Composites (ECC). In this paper, the effects of strand diameter, bond length and transverse steel strand spacing on bond behavior were studied based on 51 direct pullout tests. Experimental results showed that the high-strength stainless steel strand mesh provided specimens an excellent ductility. Based on the experimental data, the existing bond–slip model was revised using the theory of damage mechanics, which fully considered the influence of the steel strand diameter on the initial tangent stiffness of the bond–slip curve. The results of the model verification analysis show that errors are within 10% for most parameters of the bond–slip model proposed, especially in the ascending section, the errors are within 5%, indicating that the calculated results using the revised model are in good agreement with the test results. In addition, the revised model was applied to the finite element analysis by using the software ABAQUS to simulate the pullout test, in which the spring-2 nonlinear spring element was used to stimulate the bond behavior between steel strand meshes and ECC. The simulation results show that the numerical analysis fits the experimental result well, which further verifies the accuracy of the model and the feasibility and applicability of the numerical analysis method.

## 1. Introduction

Engineered Cementitious Composites (ECC) is a novel cement-based material [1,2,3]. ECC is manufactured based on the designed theory of micromechanics and fracture mechanics and is composed mainly of cement, fine-graded aggregate, admixture, and is reinforced with short fibers [4]. As a high-performance cementitious composite, the hardened composite possesses high strength, excellent ductility, and significant strain hardening behavior [5,6].

There have been studies on the applications of ECC in engineering practices [7]. In some research, ECC was used to connect beam and column, without a transverse steel bar, in which it sustained substantial shear distortions under cyclic loads [8]. Mechtcherine [9] introduced the novel material to strengthen and repair buildings and infrastructure made of concrete and masonry. He demonstrated that the strengthening of structures with ECC layers increased their resistance to dynamic, energetic loading as with earthquakes, impact, or exposures. Some scholars [10] used ECC for the flexural repair of concrete structures with significant steel corrosion and achieved an excellent reinforcement effect, which proved that an ECC patch can fully recover the load-carrying capacity of the corroded steel rebar during the hardening of the rebar.

However, as a cement-based material, ECC cannot completely replace the steel bar to bear the load in a certain direction. Researchers have proposed their use in combination with reinforcement materials for numerous structural applications [11,12]. To develop a good performance of the composite structure, the assessment of bond behavior between reinforcement materials and ECC matrix is a key aspect [13].

Xu et al. [14] proposed a bond–slip model of steel bat-concrete where the parameters, including bar diameter, anchor length, concrete strength, and cover thickness, were considered comprehensively. A modified model [15], developed by Lundgren, can be used to predict splitting failures and the loss of bond and is certified by conducting a finite element model. Some researchers have also studied the influence of concrete age on bond strength [16,17]. The results demonstrated that bond strength between concrete and steel rebar was affected by concrete age significantly and decreased with increasing curing age [18]. Based on these, Shen et al. [19] have further studied the bond behavior between steel bars and high-strength concrete at different ages and conducted a prediction model for the bond stress-slip relationship considering the effect of concrete age and concrete strength.

The bond behavior of steel bar and ECC is a hot research topic currently [20,21,22,23]. Hou [24] studied the effect of corrosion on the bond and found that corrosion scarcely weakened the bond toughness between ECC and corroded rebar below a 15% corrosion ratio. The influence of high temperature has also been investigated [25,26]. Test results proved that excellent bonding performance was built between the rebar and ECC, even at temperatures up to 800 °C.

Lee et al. [20] proposed an analytical model for the bond–slip relationship of steel bar-ECC, in which a nested iteration procedure was employed for embedded reinforcement under pull-out forces, and the effect of embedment length on the failure mode of reinforcement was studied. A more comprehensive constitutive model conducted by Zhou [27] can be adopted to predict the bond–slip behaviors of steel bar-ECC with FRP confinement, and used as a reference for application in practical engineering.

In the study of the bond behavior of FRP bars embedded in cement matrix, the researchers [28,29] found that the bond strength decreased with the increase of embedment length, and smaller diameter bars developed higher bond strengths than larger ones. A local bond–slip relationship model and the method for the determination of the parameters of the model was proposed [30]. The method can be applied to take into account different embedment lengths. Other studies were conducted to expose the effects of bonding by fiber [31,32]. The bond behavior of FRP-ECC was affected by fiber type, fiber volume fraction. PVA and hook end steel fibers were able to increase the bond behavior because they can resist and control the interfacial crack initiation, growth, and propagation.

The model describing bond–slip relationships between textiles and cementitious matrix was reported initially by Naaman et al. [33] and has been developed by other scholars [34]. Banholzer [35] proposed a model with iteration procedures that introduced the slip distribution law and boundary conditions. In further research [36], the transverse elements improved not only the bonding behavior but also the toughness of composite materials. Jiang [37] described the bond–slip curve in one continuous function that can be used to derive the bond force and proposed an equivalent method to evaluate the effect of weft yarns.

Steel strand has high strength, good fire resistance, and good economic, compared with reinforcement and textile. Currently, a combination of steel wire and cement-based material is popular in the rehabilitation of RC structures [38]. Kim [39] reported an experiment using steel wire mesh and permeable polymer mortar to retrofitted RC bridge columns and he suggested a satisfactory increase in load carrying capacity and improving substantially in the hysteretic behaviors of the column. Steel wires combined with ECC were used to improve the bearing capacity and durability of existing RC beams, in which the crack development in the concrete tension zone was delayed [40]. To develop a good performance of the composite structure, the assessment of bond behavior between steel strand and ECC matrix is a key aspect, but few relevant studies reported on this issue.

Compared with the aforementioned reinforced materials, the high-strength stainless steel strand has a large difference in both apparent morphology and mechanical properties. Whether this type of bond property can be correctly expressed by existing models still needs further investigation. As shown in Figure 1, a series of pull-out experiments were conducted and the effect of parameters including the bar diameter, the relative embedded length, and the transverse steel strand spacing were investigated. The bond performance and the characteristics of bond–slip curve were discussed on the basis of the experimental results. Afterward, on the basis of the bond–slip model previously developed, a constitutive model was proposed for stimulating the bond–slip relationship between steel strand and ECC, which considered additionally the effect of the strand diameter on the initial tangential stiffness and the damage inside ECC substrate. Furthermore, the finite element was developed to verify the validity of the bond–slip model.

## 2. Experimental Methods and Results

### 2.1. Specimen Design

In this paper, a total of 51 sets of pullout tests were conducted, which were divided into 17 groups according to different experimental parameters. Each group had three identical specimens. The parameters considered in the experiment were the strand diameter *d* (2.4 mm, 3.2 mm, and 4.5 mm), the relative embedded length *l_a_* (15*d*, 18*d*, 20*d*, 22*d*, 25*d,* and 28*d*), and the transverse steel strand spacing *l_d_* (20 mm, 30 mm and 40 mm).

As shown in Figure 2, the specimen size is *a* × *b* × *c*, where *a* indicates the specimen width (150 mm), *b* indicates the specimen length (100 mm, 150 mm, and 170 mm), and *c* indicates the specimen thickness (27 mm, 37 mm, and 50 mm). The specimens were cast in cuboid molds with a longitudinal steel strand embedded horizontally, *d*/2 mm below the central axis. Transverse steel wires are located at *d*/2 below the center of the specimen, and on the left and right end, extend 2–3 mm from the template. The longitudinal steel wires at the ends of the specimen are sheathed in PVC pipes to prevent the bonding between two materials, which can avoid the local failure of ends resulting from the stress concentration.

The standard specimens were manufactured according to the following steps: (1) fixtures were used to fix the steel strand in a predetermined position and make it in a tight state; (2) the mutually contacting parts of the steel strand in two directions were tied together by steel wires; (3) polyurethane foam sealing agent was applied to all the gaps in the template to avoid cement leakage; (4) ECC could be cast. In addition, a number of cubic specimens with sizes of 70.7 mm × 70.7 mm × 70.7 mm and 280 mm × 40 mm × 15 mm were also cast for compression and tension tests respectively. In the material properties test of ECC, the compressive strength is 32.45 MPa and the specimen exhibits a tensile strain-hardening behavior up to approximately 2.2% strain, with a tensile strength close to 2.83 Mpa and a cracking strength close to 1.89 MPa.

The specimens were numbered in the form A-B-C-D, where A denotes the diameter of the steel strand, B denotes the relative anchorage length, C denotes the spacing between transverse steel strands, and D represents the order of the specimen in the group. For example, a specimen named 4.5-15*d*-30-1 denotes that the diameter of the steel strand is 4.5 mm, the relative anchorage length is 15*d*, the spacing between transverse steel strands is 30 mm, and it is the first specimen in the group. Additional details on the experimental program are available in the literature [41].

### 2.2. Test Results

The setup for the pullout tests is shown in Figure 3. The AB segment is the free end, the CD is the loading end, and the BC segment is the actual anchoring segment of the longitudinal steel strand in the ECC. The loads were applied to the specimens by a 100 kN capacity servo-hydraulic testing machine, and were measured by the testing machine directly. Besides, four linear variable differential transducers (LVDTs) were installed to measure the slip of the steel strand relative to the cement.

Test results are given in Table 1, which includes the failure modes, the ultimate pullout force *T_a_*, the ultimate bond stress *τ**_a_*, and the corresponding displacement *s_a_*. The average bond stress and relative slip between anchorage steel strand and ECC were calculated as follows:(1)$$\tau =\frac{F}{\pi dl_{a}}$$
(2)$$s=\frac{s_{C}+s_{A}}{2},$$
where *τ* and *s* denote the average bond stress and slip respectively, *F* indicates the applied load, *d* indicates the steel strand diameter, *l_a_* indicates the anchorage length. *s_A_* denotes the slip at the free end and it can be ignored because the deformation of the AB segment steel strand is extremely small, and therefore the displacement at point A is regarded as point B. *s_C_* denotes the slip at the loading end. Because the deformation of the *CD* segment cannot be ignored, *s_C_* is calculated as follows:(3)$$s_{CD}=\frac{Fl_{CD}}{E_{s}A_{s}}$$
(4)$$s_{C}=s_{D}-s_{CD},$$
where *l_CD_* is the *CD* segment length, *E_S_* and *A_S_* are the elastic modulus and the measured area of the steel strand, *s_CD_* is the elastic deformation of the *CD* segment, and *s_C_* and *s_D_* are the displacements at point C and point D, respectively.

### 2.3. Analysis of the Pullout Process

As shown in Figure 4, the representative load-displacement curves are plotted according to the test results, and exhibit the following characteristics; (1) the ascending segment of the curve is very steep; (2) the slip at the free end was lower than the load end, which indicates that the pullout force starts from the load end and develops towards the free end; (3) with the join of transverse steel wires, the curves of those specimens show obvious ductility when the load is reduced to 80% of the maximum. Therefore, bond failure is transformed into ductile failure from brittle failure at the interface of the strand and ECC.

## 3. Bond–Slip Model

### 3.1. Bond–Slip Curve

The representative bond–slip curves are shown in Figure 5, divided into five stages: upward stage, first descending stage, ductile strengthening stage, second descending stage, and residual stage.

(1)Upward stage (OA): In the initial stage, the bond stress develops rapidly while the slip increases very limited. The bonding force is mainly provided by chemical adhesive and mechanical interlocking. As the free end begins to slip, the chemical adhesive completely disappears, but the actions of mechanical interaction and friction resistance are gradually obvious;(2)First descending stage (AB): After reaching the peak stress, the bond stress begins to decrease because of the rapid decrease of mechanical interaction force, and the increase of friction resistance is very limited. The role of the transverse steel wires is to reduce the decrease rate of the curve and thus help ECC bear the stress;(3)Ductile strengthening stage (BC): After being destroyed to a certain degree, the traverse steel strand contributes to retain the mechanical interaction. At the same time, the friction resistance is constantly increasing. Under the combined action of the two forces, the bond stress remains stable, macroscopically;(4)Second descending stage (CD): In this stage, the bond–slip curve shows a more obvious descending trend. This is because the action of mechanical interlocking and friction are both weakened gradually and the role of horizontal steel wires is also weakened further;(5)Residual stage (DF): At last, the bond–slip curve tends to be flat, and the bonding force is provided only by friction resistance. Finally, the longitudinal steel strand is pulled out.

### 3.2. Development of an Analytical Bond-Slip Model

A suitable bond–slip constitutive relationship is the key to further research and the foundation for engineering application. As a novel civil engineering material, ECC has significant differences from conventional concrete, and there are few studies on the bond–slip relationship between ECC and steel strand. In the existing model [42,43], neither of them can describe such a distinctive ductile stage. Based on the test results and existing models, a new bond–slip constitutive model is proposed for the steel strand–ECC. The mathematical expression could be written as Equation (5), and the corresponding typical bond–slip model is plotted in Figure 6.
(5)$$\tau =\left\{ \begin{array}{l} \left(1-d_{e}\right)K_{0}s\;\;\;\;\;\;\;\;\;\;\;\;\;\;\;\;\;\;\;\;\;\;\;\;\;\;\;\;\;\;\;\;\;\;\;\;\;\;\;\;\;\;\;\;\;\;0\leq s<s_{a}\\ \left[\frac{1+\varepsilon_{1}}{2}+\frac{1-\varepsilon_{1}}{2}\cos\frac{\pi \left(s-s_{a}\right)}{s_{b}-s_{a}}\right]\tau_{a}\;\;\;\;\;\;\;\;\;\;\;\;\;\;\;\;\;\;\;\;s_{a}\leq s<s_{b}\\ \varepsilon_{1}\tau_{a}\;\;\;\;\;\;\;\;\;\;\;\;\;\;\;\;\;\;\;\;\;\;\;\;\;\;\;\;\;\;\;\;\;\;\;\;\;\;\;\;\;\;\;\;\;\;\;\;\;\;\;\;\;\;\;s_{b}\leq s<s_{c}\\ \left[\frac{\varepsilon_{1}+\varepsilon_{2}}{2}+\frac{\varepsilon_{2}-\varepsilon_{2}}{2}\cos\frac{\pi \left(s-s_{c}\right)}{s_{d}-s_{c}}\right]\tau_{a}\;\;\;\;\;\;\;\;\;\;\;\;\;\;\;\;s_{c}\leq s<s_{d}\\ \varepsilon_{2}\tau_{a}\;\;\;\;\;\;\;\;\;\;\;\;\;\;\;\;\;\;\;\;\;\;\;\;\;\;\;\;\;\;\;\;\;\;\;\;\;\;\;\;\;\;\;\;\;\;\;\;\;\;\;\;\;\;\;\;\;\;\;s_{d}\leq s \end{array}\right.,$$
where *d_e_* denotes the damage evolution parameter; *K*_0_ denotes the initial stiffness of the curve; *τ_a_*, *τ_b_*, *τ_c_* and *τ_d_* (unit: MPa) indicate the bond stress of points A, B, C and D, respectively; *s_a_*, *s_b_*, *s_c_* and *s_d_* (unit: mm) indicate slip value of points A, B, C and D, respectively; *ε*_1_ and *ε*_2_ denote dimensionless coefficients, which make it more concise to express the formula.

### 3.3. Prediction of the Model Parameters

Figure 7 shows the bond–slip curves in the upward stage. The curve is significantly affected by the strand diameter, hence when formulating the curve, it is reasonable to adopt a function involving diameter. Based on damage mechanics, a model is adopted to describe the bond–slip by Mazars J et al. [42]. It is taken here to plot the upward stage of bond–slip behavior between ECC and steel strand mesh. The model includes the influence of various factors having clear physical meanings, not just the diameter. The formulation is defined in Equation (6) where *d_e_* denotes the damage evolution parameter, and *K*_0_ denotes the initial stiffness of the curve.
(6)$$\tau =\left(1-d_{e}\right)K_{0}s\;\;\;0\leq s<s_{a}$$
(7)$$K_{0}=\frac{\tau }{s}=\frac{0.8\alpha_{0}E}{d}$$
(8)$$d_{e}=1-\frac{\rho_{e}n}{n-1+{\left(s/s_{a}\right)}^{n}}$$
(9)$$\rho_{e}=\frac{\tau_{a}}{K_{0}s_{a}}$$
(10)$$n=\frac{K_{0}}{K_{0}s_{a}-\tau_{a}}.$$

Zhou et al. [27] have explained the calculation method of *K*_0_ in detail, and the final formulation could be written as Equation (7). However, there is no introduction to the calculation method of *α*_0_, which is used to denote the instantaneous deformation depth. Fitting with the experimental data, *α*_0_ is significantly affected by the strand diameter. It can be deduced as Equation (11) where *d*_3_._2_ indicates that the steel strand diameter is 3.2 mm. It is noticeable that the effect of steel strand diameter often has a limit value, and fitting of the test results denotes that *α*_0_ ≥ 0.03 is more appropriate for steel strand–ECC. The comparison between measured and predicted bond–slip curves in the upward stage is shown in Figure 8, which proves that the proposed model fits well with test data.
(11)$$\alpha_{0}=\{\begin{array}{l} {\left[\ln\left(-0.04{\left(d/d_{3.2}\right)}^{2}+0.295\left(d/d_{3.2}\right)+0.852\right)\right]}^{2}\\ \geq 0.03 \end{array}.$$

In Equation (5), *s_a_* denotes the slip value corresponding to the peak value of bond strength, which is reduced due to part of the tensile stress is transferred to the transverse steel strand in the ECC, and consequently the shear deformation of the ECC becomes smaller. In the upward stage, where no obvious shear failure takes place, the relative slippage between the two materials is reduced due to the presence of transverse steel wires. The formulation of *s_a_* could be written as Equation (12):(12)$$s_{a}=(0.002\times d+0.657)\times \left(0.018\times \frac{l_{a}}{d}+0.741\right)\times \left(-1.781\times \frac{d}{l_{d}}+1.402\right),$$
where *d* denotes the steel strand diameter, mm; *l_a_* denotes the embedded length, mm; *l**_d_* denotes the transverse steel strand spacing, mm.

The peak value of bond strength *τ_a_* can be obtained by solving the simultaneous equations of Equations (6) and (12), but when calculating *τ_a_* only, this method is cumbersome, so it is necessary to establish a concise formulation. Based on the regression analysis of the experimental data in this paper, the following equation can be deduced:(13)$$\tau_{a}=\left(-0.06\times \frac{d}{d_{3.2}}+1.242\right)\times \left(-0.019\times \frac{l_{a}}{d}+5.828\right)\times \left(0.204\times \frac{d}{l_{d}}+0.578\right)\times f_{et},$$
where *d*, *d*_3_._2_, *l_a_* and *l_b_* have already been introduced in the previous part of the paper, *f_et_* denotes the tensile strength of ECC, *f_et_* is 2.83 MPa.

The endpoint of the first descending stage *τ_b_*, as shown in Figure 6, is written as *ε*_1_*τ_a_*. When *s* > *s_a_*, without the intervention of the transverse steel strand, the bond force will be reduced to 0.5–0.6 times *τ_a_*. The addition of the transverse steel strand makes *ε*_1_ at least greater than 0.5. Based on the pull-out test result, the formula is shown as Equation (14), and the corresponding slip is shown as Equation (15):(14)$$\varepsilon_{1}=0.5+\left(-0.037\times \frac{l_{a}}{d}+6.034\right)\times \left(0.057\times \frac{d}{l_{d}}+0.062\right),$$
(15)$$s_{b}=\left(0.079\times \frac{d}{d_{3.2}}+0.111\right)\times \left(0.02\times \frac{l_{a}}{d}+2.61\right)\times \left(0.011\times l_{d}+2.394\right).$$

The length of ductile strengthening stage *s_c−b_* is mainly related to the steel strand diameter *d*, the embedded length *l_a_*, and the transverse steel strand spacing *l_d_*. Fitting with the experimental data, it is given in Equation (16).
(16)$$s_{c}-s_{b}=\left(0.188\times \frac{d}{d_{3.2}}-0.069\right)\times \left(-0.243\times \frac{l_{a}}{d}+9.26\right)\times \left(-0.216\times l_{d}+10.958\right).$$

In the same way as *τ_b_*, *τ_d_* is written as *ε*_2_*τ_a_*, and the formula is shown as Equation (17). The corresponding slip value *s_b_*, which is ideally expressed as *s_b_* = g[exp(*d*/*d*_3_._2_), *l_a_*/*d*, *l_d_*], because it is significantly affected by the steel strand diameter. The formula is shown as Equation (18).
(17)$$\varepsilon_{2}=\left(0.138\times \frac{d}{d_{3.2}}+0.072\right)\times \left(0.013\times \frac{l_{a}}{d}+0.323\right)\times \left(-0.1\times \frac{l_{a}}{l_{d}}+3.1\right),$$
(18)$$s_{d}=e^{\left(1.531\times \frac{d}{d_{3.2}}+0.257\right)}\times \left(-0.013\times \frac{l_{a}}{d}+0.941\right)\times \left(-0.013\times l_{d}+1.814\right).$$

### 3.4. Model Verification

The prediction method of the model parameter introduced in Section 3.3 is used to predict the experiment curve. The comparison results of the prediction and experiment values are all gathered in Table 2, and the mean and coefficient of variation (C.V.) of the ratio of tested and predicted values were also listed. Figure 13 compares bond–slip curves measured in this test and predicted by the proposed model that considers different parameters including steel strand diameter, bond length, and transverse steel strand spacing. As shown in Table 2 and Figure 13, the curves predicted by the proposed model are always in good agreement with the test curves, demonstrating the satisfied accuracy of the proposed model in this paper.

## 4. Finite Element Modeling

Finite element analysis has gradually developed into an extremely crucial research tool that has been used to supplement and expand experiments by many scholars in engineering, biology, medicine, and aeronautic fields specimen in the group. Additional details on the experimental program are available in the literature [44,45]. It can solve several difficulties of experiments, such as long experimental periods, discrete results, and limited research parameters. Data that are difficult to measure or phenomena that cannot be observed also can be simulated by finite element. In this study, the general-purpose finite element package ABAQUS was used to check the effectiveness of the experimentally assessed relationship in simulating the bond behavior of ECC and steel strand by establishing a three-dimensional finite element. The steel stand and ECC are modeled using solid and beam elements, respectively. A nonlinear spring element is used to simulate the bond between steel strand and ECC, which is commonly adopted in literature for such a type of analysis.

### 4.1. Finite Element Model Geometry

Three-dimensional (3D_ full-scale models were created to simulate the pullout test, and these models are taken non-linear finite element analysis in order to better understand the bond behavior between ECC and steel strand mesh. A typical numerical model and the corresponding boundary conditions are plotted in Figure 9. The steel strand and ECC are modeled separately. ECC is established as a solid element, and the element type is C3D8R, the first-order, reduced integration hexahedral continuum element. Steel strand is established as a wire element due to the big aspect ratio, and the element type is B31, the 2-node linear beam in space.

The spring element is used to describe the force-displacement relationship between two nodes, and simulate the bond behavior between two materials, macroscopically. In ABAQUS, there are three types of spring elements: Spring-1, Spring-2, and Spring-A. Spring-2 element, all of which are imaginary mechanical models with only stiffness and no actual geometric dimensions and mass. Spring-2 element is adopted to simulate the bond behaviors between steel stand and ECC and it is a nonlinear spring element, including transverse stiffness and tangential stiffness. The methods to determine stiffness values are discussed in Section 4.2.

As shown in Figure 9, the identical constraint conditions of the experimental specimens were setup in the numerical model. The vertical displacement was confined to the face of concrete at the loaded end, and the other faces of the ECC specimen were left free of displacement constraints. The loading condition was a displacement to be put to the bottom node of the steel strand in the pullout direction.

### 4.2. Material Models, Properties, and Parameters

In ABAQUS, there is no standard method to simulate the material properties of ECC. As a cement-based composite material, ECC can be simulated by the same method as concrete, based on the reason that they share similar characteristics. Under low-pressure stress, the models adopted to simulate concrete properties include Concrete Smeared Cracking model, Concrete Brittle Cracking model, and Concrete Damage Plasticity model. In this paper, the Concrete Damage Plasticity (CDP) model is used to simulate the mechanical behavior of the ECC, and CDP is characterized by three sections: plasticity, compressive behavior, and tensile behavior. For the plasticity section, there are five parameters be defined: the dilation angle (*ψ*), eccentricity parameter (*ε*), the ratio of the biaxial compressive strength to the uniaxial compressive strength (*f_b_*_0_/*f_c_*_0_), the ratio of the second invariant of the deviatoric stress tensor in tensile meridians to that in compressive meridians (*K_C_*), and Viscosity Coefficient (VC). These parameters are obtained by reference to available literatures and contrast trial calculation results. A set of data is accepted based on the numerical results that match the experimental or theoretical results with a minimum average error. The final values selected are listed in Table 3.

In the material property test, the elastic modulus and Poisson’s ratio of ECC are 14.5 GPa and 0.255, respectively. For the compressive behavior of ECC, the uniaxial compressive stress-strain curve is shown in Figure 10a and can be expressed by:(19)$$\frac{\sigma }{\sigma_{cu}}=\{\begin{array}{l} \;1.1\frac{\varepsilon }{\varepsilon_{cu}}+0.5{\left(\frac{\varepsilon }{\varepsilon_{cu}}\right)}^{5}-0.6{\left(\frac{\varepsilon }{\varepsilon_{cu}}\right)}^{6}\;\;\;0\leq \frac{\varepsilon }{\varepsilon_{u}}<1\\ \;\frac{0.15{\left(\frac{\varepsilon }{\varepsilon_{cu}}\right)}^{2}}{1-2\frac{\varepsilon }{\varepsilon_{cu}}+1.15{\left(\frac{\varepsilon }{\varepsilon_{cu}}\right)}^{2}}\;\;\;\;\;\;\;\;\;\;\;\;\;\;\;\;\;\;\;\;\;\;\;\;\;\;\frac{\varepsilon }{\varepsilon_{cu}}\geq 1 \end{array},$$
where *σ* and *ε* denote the compressive strain and stress of ECC, respectively; *σ_cu_* and *ε_cu_* denote the maximal compressive strength (peak point of the curve) and strain of ECC, respectively; *σ_cu_* is 32.45 MPa, and *ε_cu_* is 0.022.

The uniaxial tensile stress-strain curve of ECC is shown in Figure 10b and can be expressed by the following equation:(20)$$\sigma =\{\begin{array}{l} E_{c0}\varepsilon \;\;\;\;\;\;\;\;\;\;\;\;\;\;\;\;\;\;\;\;\;\;\;\;\;\;\;\;\;\;\;\;\varepsilon \leq \varepsilon_{tc}\\ (0.31\frac{\varepsilon }{\varepsilon_{tu}}+0.69)\sigma_{tu}\;\;\;\;\;\;\varepsilon_{tc}<\varepsilon \leq \varepsilon_{tu} \end{array},$$
where *E_c_*_0_ denotes the elastic modulus of ECC, MPa; *σ* and *ε* denote the tensile stress and strain of ECC, respectively; *σ_tu_* and *ε_t_**_u_* denote the maximal tensile strength (peak point of the curve) and corresponding strain, respectively, *σ_tc_* and *ε_tc_* denote the nominal tensile cracking stress and strain of ECC; *σ_tc_* is 2.83 MPa, and *ε_tc_* is 1.85 × 10^−4^; *σ_tu_* is 3.86 MPa, and *ε_tu_* is 0.0271.

Longitudinal reinforcement was modeled using the elasticity plasticity model with isotropic strain hardening. The measured tensile properties are reported in Table 4. For steel strand under uniaxial tensile loading, the typical constitutive relationship is shown in Figure 11, which can be expressed as Equation (21), according to different steel strand diameters.
(21)$$\sigma =\{\begin{array}{l} E_{s}\varepsilon_{s}\;\;\;\;\;\;\;\;\;\;\;\;\;\;\;\;\;\;\;\;\;\;\;\;\;\;\;\;\;\;\;\;\;\;\;\varepsilon \leq \varepsilon_{y}\\ a(\frac{\varepsilon }{\varepsilon_{u}}{)}^{3}+b(\frac{\varepsilon }{\varepsilon_{u}}{)}^{2}+c\frac{\varepsilon }{\varepsilon_{u}}\;\;\;\varepsilon_{y}<\varepsilon \leq \varepsilon_{u} \end{array},$$
where *E_s_* denotes the elastic modulus of steel strand; *ε_y_* denotes the strain of steel strand, the plasticity beginning; *σ* and *ε* denote the tensile stress and strain of steel strand, respectively; *σ_u_* and *ε_u_* denote the maximal tensile strength and corresponding strain, respectively; a, b and c are dimensionless coefficients. For all these parameters, their values are obtained by fitting the test results and listed in Table 4.

The Spring-2 elements are established on the nodes where the ECC and steel strand coincide. The number and spacing of the non-linear spring elements are determined by the bond length of the steel strand. It is worth noting that to ensure the nodes of the steel strand element coincide with the nodes of ECC element, by selecting the appropriate element length. In the FE model, spring elements are established in normal and tangential directions, and are used to simulate the squeeze between the steel strand and ECC and the slip between the two materials, respectively. The stiffness of normal spring is taken as a constant value, which is the same as the elastic modulus of ECC, which is 14.5 GPa. The stiffness of the tangential spring is obtained through the conversion of the bond–slip relationship obtained through experiments, which can be expressed as follows:(22)$${Κ}_{t}=\frac{d\tau }{ds}\times \pi dl_{a}.$$

According to the correspondence between *τ* and *s* in Equation (5), the tangential stiffness *K_t_* is obtained by combining Equations (5) and (22). Thus, the constitutive relation between spring element force and node displacement difference as expressed in Equation (23) can be determined.
(23)$$\left[\begin{array}{c} F_{t}\\ F_{v1}\\ F_{v2} \end{array}\right]=\left[\begin{array}{ccc} K_{t} & 0 & 0\\ 0 & K_{v1} & 0\\ 0 & 0 & K_{v2} \end{array}\right]\left[\begin{array}{c} \Delta u_{t}\\ \Delta u_{v1}\\ \Delta u_{v2} \end{array}\right],$$
where *K_t_*, *K_v_*_1_ and *K_v_*_2_ are tangential spring stiffness and two normal spring stiffness, *F_t_*, *F_v_*_1_ and *F_v_*_2_ denote the corresponding spring forces, ∆*u_t_*, ∆*u_v_*_1_ and ∆*u_v_*_2_ denote the corresponding displacement differences, respectively.

### 4.3. Comparison between Numerical and Experimental Results

As shown in Figure 12, the curve of the load-displacement relationship predicted by the proposed FE model is compared with the corresponding experimental results. It can be observed that the computed result fits the experimental result quite well when the curve is in the upward stage. In the latter part of a curve, the difference between the two curves is mainly due to the manufacturing error of the specimens. The comparison of the bond–slip curve is presented in Figure 13, in which the experimental curve, numerical curve, and stimulated curve are all plotted. In all comparison figures, the little difference is acceptable in highly nonlinear ranges of behavior; therefore, it proves that the developed model is accurate.

However, this model is still based on the average bond–slip relationship. As shown in Figure 4, there is a difference between the curves at the free end and loaded end. The study of local bond–slip relations may be carried out subsequently, and the entire bond–slip process will be carried out to analyze the real situation of the interface.

## 5. Conclusions

In this paper, experiments on seventeen groups of specimens with or without the addition of traverse steel stand were conducted to investigate the bond behavior of the steel strand–ECC system. For specimens with the traverse steel strand, the bond–slip curve showed an obvious ductility stage because the destruction speed of mechanical interlocking was reduced significantly with the part of stress in ECC borne by the traverse steel strand.

Based on the test data, a constitutive formula was proposed to predict the bond–slip relationship curve between the steel strand and ECC. The errors between the prediction and experiment values are within 10% for most parameters of the bond–slip model proposed, indicating the accuracy of the model. In addition, by introducing non-linear springs to model the bond behavior at the strand–ECC interface, the finite element models were established to stimulate the pull-out tests. In view of the results, the finite element model was able to simulate the bond behavior, which helps to develop a further understanding of the bar–concrete interaction during the pullout.

## Figures and Tables

**Figure 1 materials-15-05649-f001:**
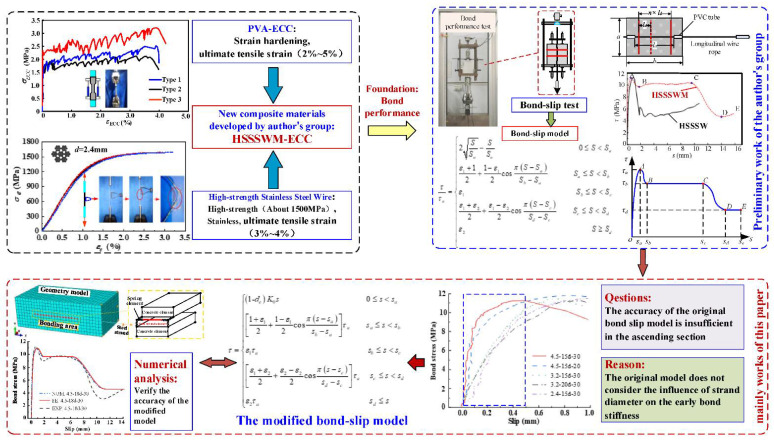
The workflow diagram.

**Figure 2 materials-15-05649-f002:**
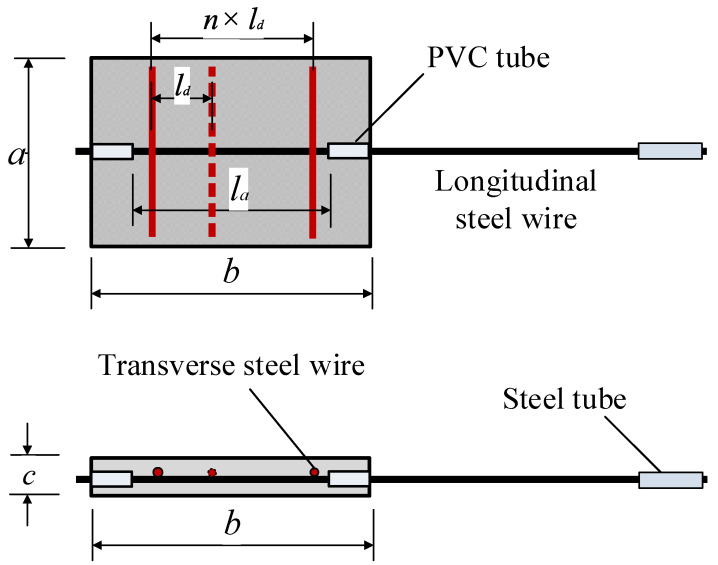
The diagram of specimens.

**Figure 3 materials-15-05649-f003:**
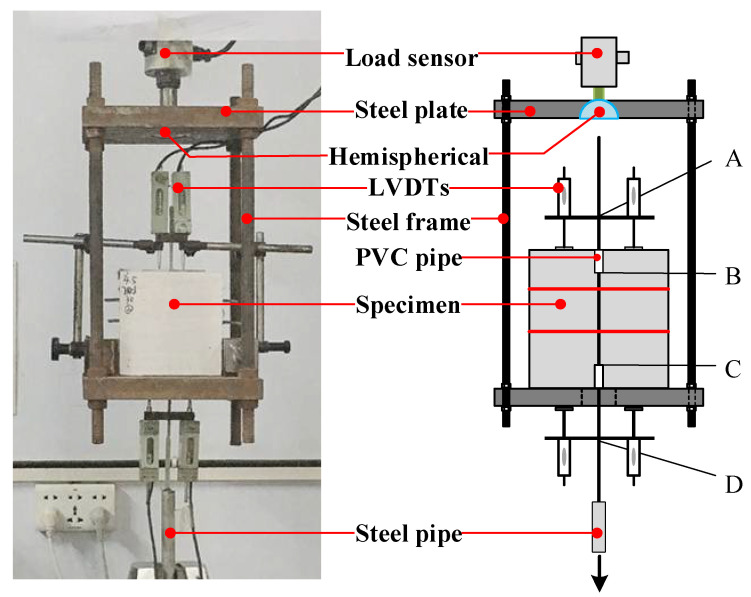
Schematic diagram of loading device.

**Figure 4 materials-15-05649-f004:**
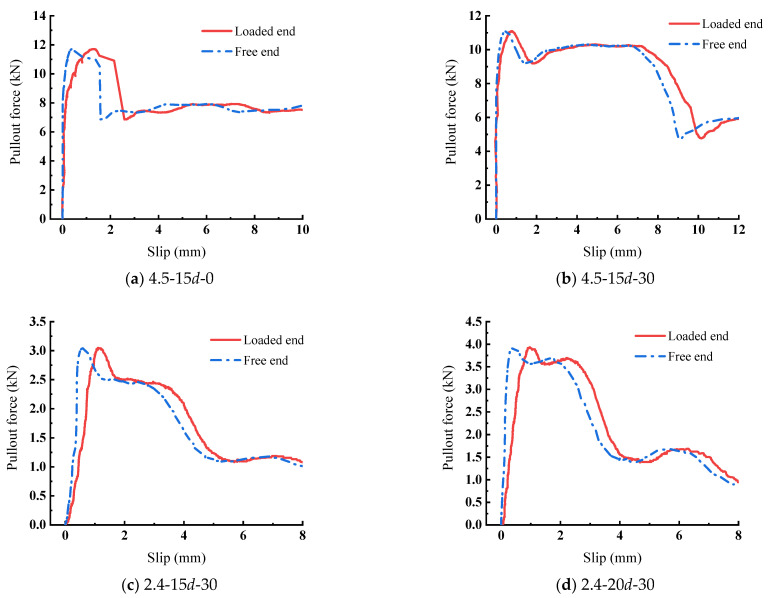
Representative pullout forces-slip curves.

**Figure 5 materials-15-05649-f005:**
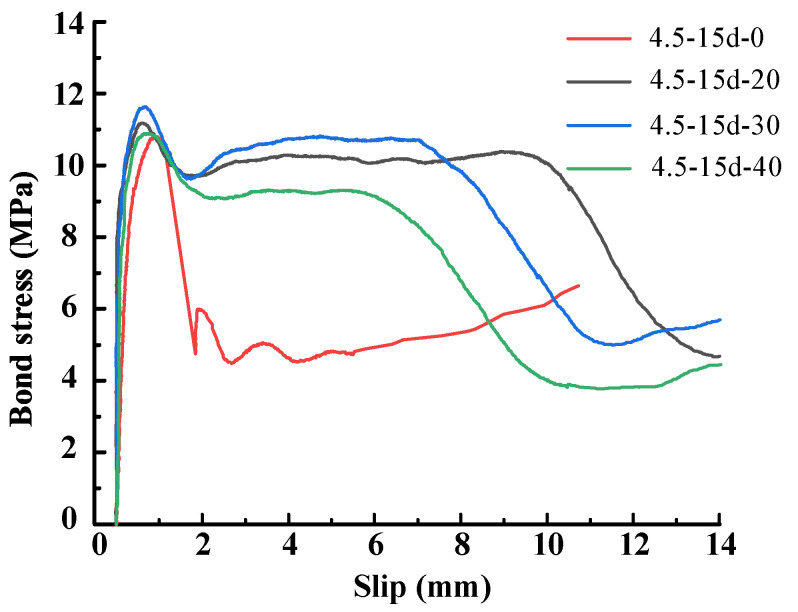
Representative bond stress-slip curves.

**Figure 6 materials-15-05649-f006:**
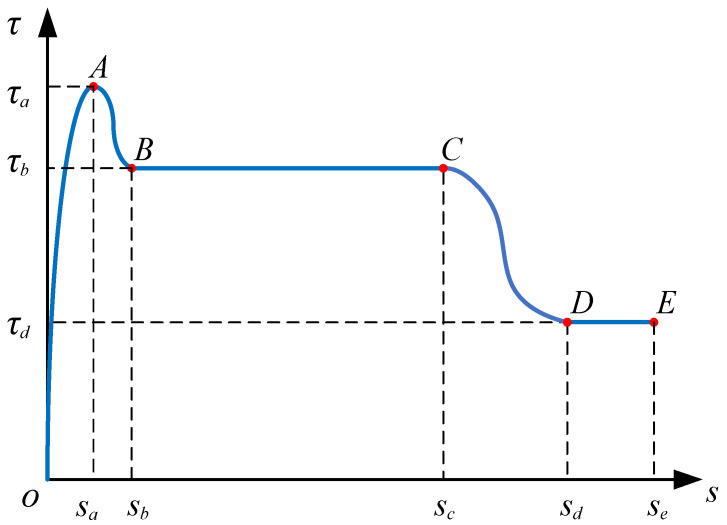
The typical bond–slip model for steel strand mesh and ECC.

**Figure 7 materials-15-05649-f007:**
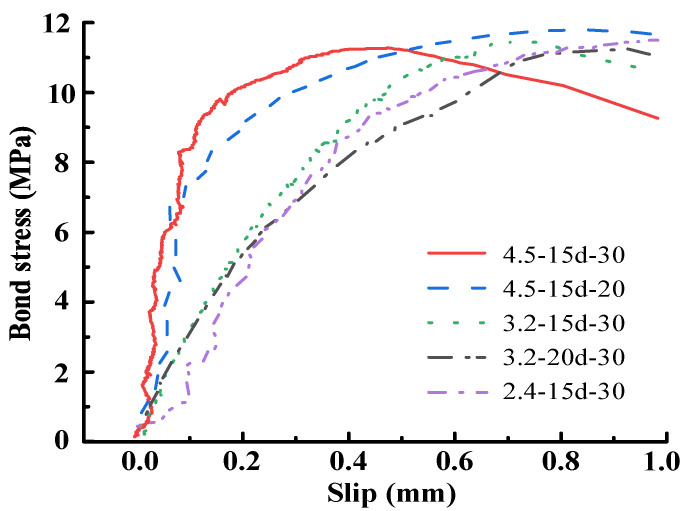
Bond slip curves in upward stage.

**Figure 8 materials-15-05649-f008:**
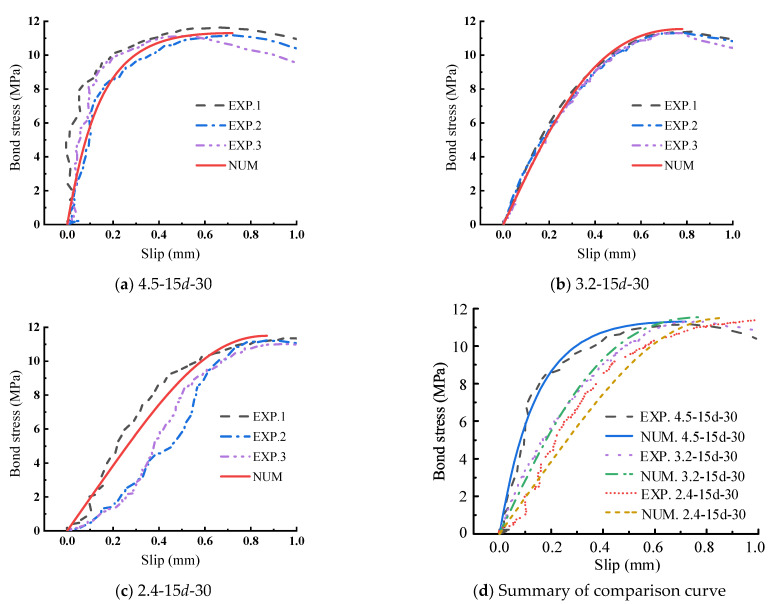
Comparison between measured and predicted bond slip curves in upward stage.

**Figure 9 materials-15-05649-f009:**
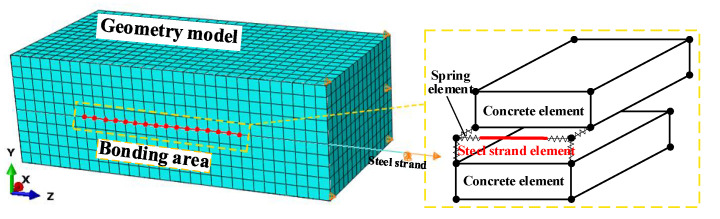
Typical finite element model.

**Figure 10 materials-15-05649-f010:**
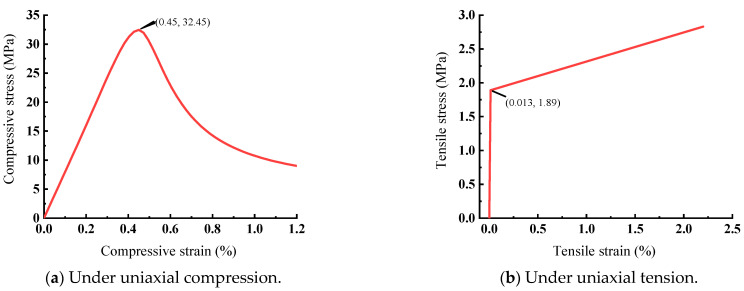
Constitutive relationships of ECC.

**Figure 11 materials-15-05649-f011:**
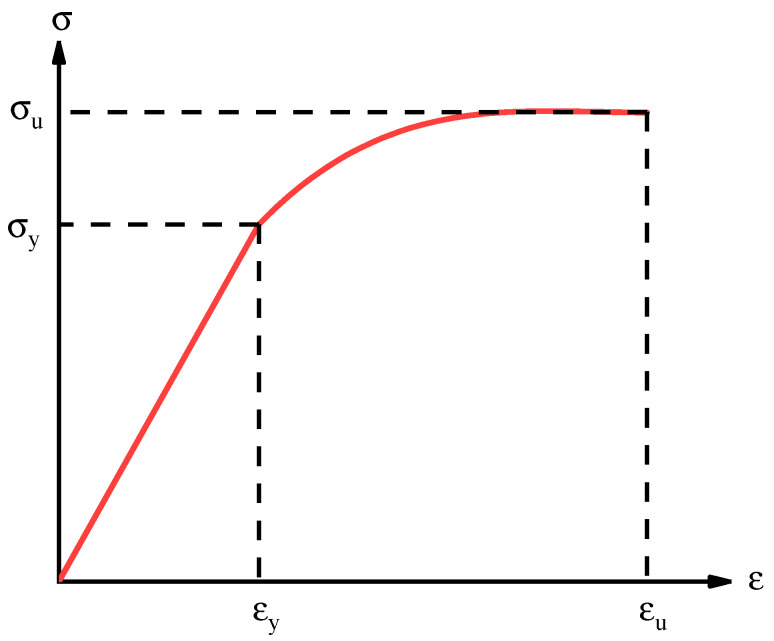
Typical constitutive relationships of steel strand.

**Figure 12 materials-15-05649-f012:**
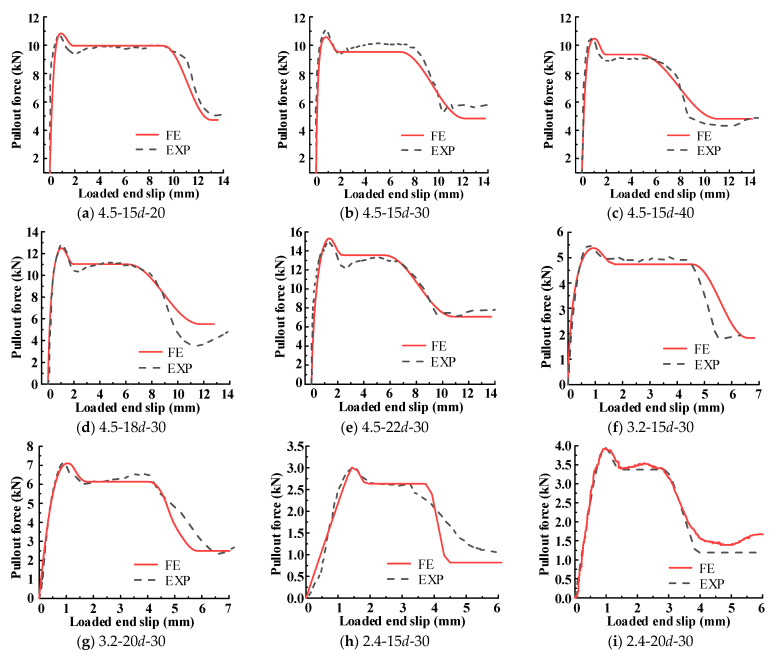
Comparison of force slip curves between simulation and test.

**Figure 13 materials-15-05649-f013:**
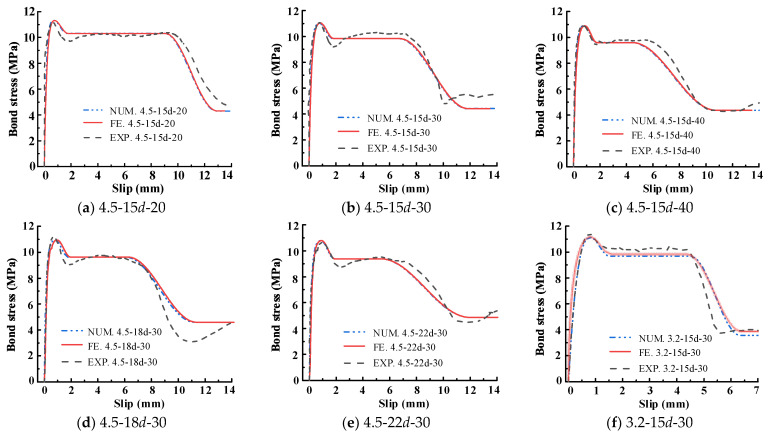
Comparison of bond slip curves of numerical calculation, simulation and test.

**Table 1 materials-15-05649-t001:** Summary of results from steel strand–ECC unidirectional pullout tests.

Group No.	Specimen No.	Specimen Size (mm × mm × mm)	*T*_a_/kN	*τ*_a_/MPa	*s*_a_/mm	Failure Mode
A1	4.5-15*d*-0-1	150 × 150 × 50	10.74	11.32	0.91	P
	4.5-15*d*-0-2		10.92	11.37	0.92	P
	4.5-15*d*-0-3		10.80	11.28	0.91	P
A2	4.5-15*d*-20-1	150 × 150 ×50	11.14	11.68	0.82	P
	4.5-15*d*-20-2		10.65	11.18	0.61	P
	4.5-15*d*-20-3		10.74	11.53	0.69	P
A3	4.5-15*d*-30-1	150 × 150 × 50	11.10	11.63	0.72	P
	4.5-15*d*-30-2		10.65	11.17	0.75	P
	4.5-15*d*-30-3		10.63	11.14	0.72	P
A4	4.5-15*d*-40-1	150 × 150 × 50	10.38	10.89	0.76	P
	4.5-15*d*-40-2		10.14	10.85	0.87	P
	4.5-15*d*-40-3		10.64	11.16	0.91	P
B1	4.5-18*d*-30-1	150 × 150 × 50	12.11	11.37	0.75	P
	4.5-18*d*-30-2		12.84	11.22	0.74	P
	4.5-18*d*-30-3		12.75	11.14	2.08	P
B2	4.5-20*d*-30-1	150 × 150 × 50	12.81	10.07	1.39	P
	4.5-20*d*-30-2		13.49	10.61	0.78	P
	4.5-20*d*-30-3		13.56	10.99	0.75	P
B3	4.5-22*d*-30-1	150 × 150 × 50	14.11	10.09	0.82	P
	4.5-22*d*-30-2		14.21	10.16	0.65	P
	4.5-22*d*-30-3		14.89	10.64	0.96	P
B4	4.5-25*d*-30-1	150 × 170 × 50	14.93	9.44	0.98	P
	4.5-25*d*-30-2		16.24	10.09	0.98	R
	4.5-25*d*-30-3		15.03	10.22	1.01	P
B5	4.5-28*d*-30-1	150 × 170 × 50	16.18	9.09	1.47	R
	4.5-28*d*-30-2		16.34	9.18	1.55	R
	4.5-28*d*-30-3		16.26	9.13	1.63	R
C1	3.2-15*d*-40-1	150 × 100 × 37	5.49	11.38	0.78	P
	3.2-15*d*-40-2		5.46	11.31	0.72	P
	3.2-15*d*-40-3		5.45	11.30	0.8	P
C2	3.2-18*d*-30-1	150 × 100 × 37	6.47	11.17	0.84	P
	3.2-18*d*-30-2		6.39	11.05	0.77	P
	3.2-18*d*-30-3		5.37	9.28	0.58	P
C3	3.2-20*d*-30-1	150 × 100 × 37	7.08	11.11	0.93	P
	3.2-20*d*-30-2		7.59	11.79	1.04	R
	3.2-20*d*-30-3		7.07	11.09	0.92	P
C4	3.2-22*d*-30-1	150 × 100 × 37	7.56	10.75	1.02	R
	3.2-22*d*-30-2		7.77	11.01	1.07	R
	3.2-22*d*-30-3		7.86	11.17	1.05	R
*D*1	2.4-15*d*-30-1	150 × 100 × 37	3.19	11.72	1.11	P
	2.4-15*d*-30-2		3.08	11.22	0.85	P
	2.4-15*d*-30-3		3.04	11.03	0.93	P
*D*2	2.4-18*d*-30-1	150 × 100 × 37	3.73	11.25	0.87	P
	2.4-18*d*-30-2		3.67	11.08	0.82	P
	2.4-18*d*-30-3		3.72	11.21	1.10	P
*D*3	2.4-20*d*-30-1	150 × 100 × 37	3.94	10.88	0.91	P
	2.4-20*d*-30-2		3.97	10.97	1.12	P
	2.4-20*d*-30-3		4.05	11.19	1.26	P
*D*4	2.4-22*d*-30-1	150 × 100 × 37	4.37	10.94	1.16	R
	2.4-22*d*-30-2		4.34	10.87	1.15	R
	2.4-22*d*-30-3		4.37	10.94	1.16	R

**Table 2 materials-15-05649-t002:** Comparison between measured and predicted parameters.

Specimen	*τ* _a,EXP_	*τ* _a,NUM_	*τ*_a,EXP_/ *τ*_a,NUM_	*s* _a,EXP_	*s* _a,NUM_	*s*_a,EXP_/ *s*_a,NUM_	*ε* _1,EXP_	*ε* _1,NUM_	*ε*_1,EXP_/ *ε*_1,NUM_	*s* _b,EXP_	*s* _b,NUM_	*s*_b,EXP_/ *s*_b,NUM_	*s* _c-b,EXP_	*s* _c-b,NUM_	*s*_c-b,EXP_/ *s*_c-b,NUM_	*ε* _2,EXP_	*ε* _2,NUM_	*ε*_2,EXP_/ *ε*_2,NUM_	*s_d_* _,EXP_	*s_d_* _,NUM_	*s_d_*_,EXP_/ *s_d_*_,NUM_
4.5-15*d*-20	11.46	11.33	1.012	0.71	0.67	1.05	0.89	0.91	0.97	1.77	1.69	1.046	7.08	7.28	0.973	0.45	0.38	1.182	12.99	12.91	1.01
4.5-15*d*-30	11.31	11.05	1.024	0.73	0.76	0.96	0.88	0.89	1.00	1.81	1.76	1.030	4.56	4.91	0.928	0.33	0.40	0.833	12.13	11.83	1.03
4.5-15*d*-40	10.97	10.91	1.005	0.85	0.81	1.05	0.81	0.88	0.93	1.91	1.83	1.041	2.33	2.54	0.916	0.4	0.40	1.000	12.53	10.75	1.17
4.5-18*d*-30	11.30	10.94	1.033	0.75	0.80	0.93	0.91	0.88	1.03	1.80	1.80	0.999	4.92	4.28	1.150	0.35	0.42	0.835	10.43	11.21	0.93
4.5-20*d*-30	10.45	10.86	0.962	0.77	0.83	0.92	0.88	0.87	1.00	1.78	1.82	0.977	3.62	3.85	0.940	0.47	0.43	1.082	11.12	10.80	1.03
4.5-25*d*-30	9.77	10.67	0.915	1.00	0.90	1.11	0.83	0.86	0.96	1.85	1.88	0.981	2.59	2.79	0.929	0.33	0.47	0.702	9.97	9.77	1.02
3.2-15*d*-30	11.33	11.12	1.019	0.77	0.81	0.94	0.89	0.87	1.02	1.34	1.51	0.890	3.48	2.99	1.164	0.37	0.32	1.157	6.18	6.35	0.97
3.2-18*d*-30	11.11	11.01	1.009	0.81	0.86	0.94	0.92	0.87	1.06	1.43	1.54	0.930	2.60	2.60	0.997	0.33	0.34	0.970	5.43	6.02	0.90
3.2-20*d*-30	11.10	10.93	1.016	0.93	0.89	1.04	0.89	0.86	1.03	1.55	1.56	0.995	2.36	2.35	1.004	0.34	0.35	0.962	5.11	5.80	0.88
2.4-15*d*-30	11.21	11.16	1.005	0.81	0.84	0.97	0.79	0.87	0.91	1.35	1.35	1.000	1.46	1.81	0.806	0.27	0.27	0.997	4.60	4.33	1.06
2.4-20*d*-30	11.01	10.97	1.004	1.02	0.92	1.12	0.93	0.85	1.09	1.56	1.40	1.115	1.16	1.42	0.815	0.39	0.30	1.296	6.12	3.95	1.55
AVE			1.000			1.001			1.001			1.004			1.001			1.003			1.000
COV			0.033			0.073			0.054			0.060			0.094			0.038			0.083

**Table 3 materials-15-05649-t003:** Plasticity of ECC input data.

*ψ*	*ε*	*f*_b0_/*f*_c0_	*K_C_*	VC
36	0.1	1.05	0.667	0

**Table 4 materials-15-05649-t004:** List of constitutive relationship parameters.

*d*/mm	*E_s_*/GPa	*ε_y_*	*ε_u_*	a	b	c
2.4	130	0.0074	0.0307	1.33	−3.66	3.33
3.2	97	0.0098	0.0408	1.45	−3.52	3.25
4.5	108	0.0076	0.0378	0.90	−2.78	2.90

## Data Availability

The data presented in this study are available on request from the corresponding author.

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
