# Peer review of "Analytical and Numerical Modeling of the Pullout Behavior between High-Strength Stainless Steel Wire Mesh and ECC"

_materials, 2022, doi:10.3390/ma15165649_

Round 1

Reviewer 1 Report

In this article, the authors explored the interaction between high-strength stainless steel wire and concrete during pullout tests. For an analytical description of this phenomenon, the authors proposed an original analytical model. A modified model was also used in the ABAQUS system. The reliability of the analytical and numerical models is confirmed by good agreement between theoretical and experimental data. The article is well written and its results are of both scientific and practical interest. The reviewer recommends the article for publication after addressing the following minor concerns.

1. The term ECC occurs in the title and in the abstract. In the abstract, you must also give the full name, and only then the abbreviation. In the title can be replaced by more general terms "bar" "concrete"?, but this is left to the decision of the authors.

2. It would also be possible and interesting to combine identical graphs in Fig. 2, 11, 12 on one. Not to replace but as additional information showing the general trend for all cases. Of course, we are talking only about the experiment, and not about the calculation.

3. Authors should carefully read the article again and distribute the information according to the relevant sections. For example, in the introduction, the authors write that the model they have developed works well. But it should be in discussion and results. Part "Introduction" should end with the statement of the main goals and objectives of the study, etc.

Author Response

Changes list and responses

Manuscript ID: materials-1855505

Title: Analytical and Numerical Modeling of the Pullout Behavior Between High-strength Stainless Steel Wire Mesh and ECC

Authors: Xu-Yan Zou, Ya-Wen Liu, Jun-Tao Zhu, Ke Li and Jing-Long Cao

The authors sincerely appreciate careful comments of reviewers and editors, which are not only for the improvement of this paper but also bring many insights to the future development of the technique. We have revised the manuscript in accordance with the comments and marked all the amends on our revised manuscript. The following is changes list and responses.

  1. The term ECC occurs in the title and in the abstract. In the abstract, you must also give the full name, and only then the abbreviation. In the title can be replaced by more general terms "bar" "concrete"?, but this is left to the decision of the authors.

Thanks the reviewer so much for the good advice. We apologize for the neglected abbreviation. We have added the full name of ECC at its first appearance in the abstract, seen in the revised paper. In addition, as described in the introduction, ECC is a high-performance cementitious composite with excellent ductility and significant strain hardening behavior. Steel strand is wound by seven bundles of seven individual high-strength steel wires. And it possesses high-strength, corrosion-resistant compared to steel bar. Therefore, it is not appropriate to replace "high-strength stainless steel wire mesh" and "ECC" in the title with the general terms "bar" and "concrete", due to the differences in their composition and mechanical properties.

  1. It would also be possible and interesting to combine identical graphs in Fig. 2, 11, 12 on one. Not to replace but as additional information showing the general trend for all cases. Of course, we are talking only about the experiment, and not about the calculation.

Thanks the reviewer's careful work for giving us the good suggestion. As shown in this paper, in order to study the bond behavior between high-strength stainless steel strand mesh and ECC, 17 groups of 51 specimens were studied by direct pullout test. Therefore, there are many test conditions, so in order to make the curve can be clearly displayed, single factor comparative analysis is adopted in the discussion. In addition, in the numerical analysis, in order to verify the accuracy of the model, the curves of different working conditions were analyzed separately, so that the curves in the figure are more clear. We tried our best to combine figures. Unfortunately, due to the excessive lines, if summed up in one picture may not be able to ensure that all lines can be clearly expressed.

  1. Authors should carefully read the article again and distribute the information according to the relevant sections. For example, in the introduction, the authors write that the model they have developed works well. But it should be in discussion and results. Part "Introduction" should end with the statement of the main goals and objectives of the study, etc.

Thanks the reviewer very much that giving us so important suggestion. We have carefully checked the article for such inappropriate statements and have changed them throughout the manuscript to be as accurate as possible. Such as, line 13-17 of page 3.

Reviewer 2 Report

1.      The title needs to be revised in uppercase and lowercase used based on MDPI format.

2.      Please provide all of the author's emails after affiliation with the initial name.

3.      Please, give quantitative results in the abstract section rather than only qualitative results.

4.      Reorder keywords based on alphabetical order.

5.      Used lowercase for each of the keywords.

6.      What is the novel of the present study? The works seem not to bring something really new and do not deliver cutting-edge insight. The authors need to clarify this issue and highlight more advance their novelty in the introduction section.

7.      The authors need to give an additional figure that presents of research workflow to make the reader more interested and easier to understand.

8.      In the present study, the authors conduct analytical and numerical studies. Why not perform experimental too?. It is encouraged to perform experimental works.

9.      The authors need to explain the advantage of using computational study rather than analytical and experimental in the finite element modeling section. Also, the suggested reference published by MDPI is needed to be adopted as follows: Computational Contact Pressure Prediction of CoCrMo, SS 316L, and Ti6Al4V Femoral Head against UHMWPE Acetabular Cup under Gait Cycle. J. Funct. Biomater. 2022, 13, 64. https://doi.org/10.3390/jfb13020064

10.   Limitations of the present study should be mentioned before the conclusion section.

11.   The conclusion section is not solid, further elaboration is needed, also make it into paragraphs, not point by point as in the present form.

12.   Reference needs to be enriched from literature published in the last five years to show the recent study. Literature from MDPI is strongly recommended.

13.   The authors need to solve the grammatical issue and improve their language style. Further proofreading is needed. Alternatively, the MDPI English editing service would be used.

Author Response

Changes list and responses

Manuscript ID: materials-1855505

Title: Analytical and Numerical Modeling of the Pullout Behavior Between High-strength Stainless Steel Wire Mesh and ECC

Authors: Xu-Yan Zou, Ya-Wen Liu, Jun-Tao Zhu, Ke Li and Jing-Long Cao

The authors sincerely appreciate careful comments of reviewers and editors, which are not only for the improvement of this paper but also bring many insights to the future development of the technique. We have revised the manuscript in accordance with the comments and marked all the amends on our revised manuscript. The following is changes list and responses.

  1. The title needs to be revised in uppercase and lowercase used based on MDPI

Thanks the reviewer's so careful work. We have corrected the title to meet MDPI format, seen the title in the revised paper “Analytical and Numerical Modeling of the Pullout Behavior Between High-strength Stainless Steel Wire Mesh and ECC”.

  1. Please provide all of the author's emails after affiliation with the initial name.

Thanks a lot for pointing out these detail questions. The emails of all the authors are now added after affiliation.

  1. Please give quantitative results in the abstract section rather than only qualitative results.

Thanks the reviewer very much that giving us so important suggestion. We have revised the abstract to be more precise. The purpose of this study is to propose a more accurate bond-slip model between high-strength stainless steel wire mesh and ECC, which can be suitably used for numerical simulation. Therefore, a quantitative description of the accuracy of the numerical simulation is added to the abstract, seen the abstract in revised paper.

  1. Reorder keywords based on alphabetical order.

Thanks a lot for pointing out these detail questions. We have reordered the keywords in alphabetical order, seen the keywords in revised paper.

  1. Used lowercase for each of the keywords.

Thanks the reviewer's so careful work. We have changed the first letter of the keyword to lowercase, seen the keywords in revised paper.

  1. What is the novel of the present study? The works seem not to bring something really new and do not deliver cutting-edge insight. The authors need to clarify this issue and highlight more advance their novelty in the introduction section.

Thanks a lot for the reviewer's important suggestion. The innovation of this paper is an improved bond-slip model between high-strength stainless steel wire mesh and ECC. As indicated in the introduction section, the new composite material high-strength steel wire mesh reinforced ECC has great potential in strengthening existing structures, and the bond performance is the basis to ensure the reinforcement efficiency. Our group has already established the corresponding bond-slip model. However, the effect of the strand diameter on the initial tangential stiffness and the damage inside ECC substrate were not considered in the preliminary model. Thus, the model is modified based on the theory of damage mechanics in this paper, and the accuracy of the model is further demonstrated on the basis of finite element numerical simulations. The statement of the innovation work in this paper has been added in the last paragraph of the introduction, seen the revised paper.

  1. The authors need to give an additional figure that presents of research workflow to make the reader more interested and easier to understand.

This is an excellent idea and thanks a lot for the reviewer's good suggestion. We have included a figure in the manuscript outlining the workflow, seen Fig. 1 in the revised paper.

  1. In the present study, the authors conduct analytical and numerical studies. Why not perform experimental too?. It is encouraged to perform experimental works.

Thanks the reviewer so much for the good advice. In fact, the author's research group has carried out relevant experimental research, and the detailed discussion of the experimental part can be found in the reference [41]: J. Zhu, K. Zhang, X. Wang, K. Li, X. Zou, H. Feng, Bond-Slip Performance between High-Strength Steel Wire Rope Meshes and Engineered Cementitious Composites [J]. Journal of Materials in Civil Engineering, 2022, 34(5): 04022048. This paper mainly revised the bond-slip model based on the analysis of test results and discusses the feasibility and accuracy of its numerical analysis.

  1. The authors need to explain the advantage of using computational study rather than analytical and experimental in the finite element modeling section. Also, the suggested reference published by MDPI is needed to be adopted as follows: Computational Contact Pressure Prediction of CoCrMo, SS 316L, and Ti6Al4V Femoral Head against UHMWPE Acetabular Cup under Gait Cycle. J. Funct. Biomater. 2022, 13, 64. https://doi.org/10.3390/jfb13020064

Thank the reviewer for these positive and detailed comments. We have rewritten to explain the advantage of finite element simulation at line 285-288 of page 7 in the revised paper. The relevant references has been cited, seen in the revised paper: references [44] and [45].

  1. Limitations of the present study should be mentioned before the conclusion section.

Thanks a lot for the reviewer's important suggestion. We have added and placed the limitations of the study at the line 387-390 of page 9 in the revised paper.

  1. The conclusion section is not solid, further elaboration is needed, also make it into paragraphs, not point by point as in the present form.

Thank you for this valuable and helpful comment, which will definitely improve our paper and make a clearer conclusion for our readers. The Conclusion section has been rewritten to give a clear and succinct conclusion in the revised paper.

  1. Reference needs to be enriched from literature published in the last five years to show the recent study. Literature from MDPI is strongly recommended.

Thanks so much for your useful comments. The references have been updated in the revised paper.

  1. The authors need to solve the grammatical issue and improve their language style. Further proofreading is needed. Alternatively, the MDPI English editing service would be used.

I appreciate very much that the reviewer read this article so carefully. I have checked the paper carefully for several times, if the reviewer feels that the grammar is still not standard enough, the author will be to accept the help of MDPI's English editing service.

Reviewer 3 Report

In this paper, the effects of rebar diameter, bond length and transverse 11 steel strand spacing on bond behavior were studied based on 51 direct pullout tests. Experimental 12 results showed that the high-strength stainless steel strand mesh provided specimens an excellent 13 ductility. Following points must be addressed to proceed for publication

1.  In Abstract, the digital values of thermal measurement should be presented, absorbing the attention of readers. At present, the Abstract is declarative sentences, while the outcomes obtained in this work are lost, especially for the digital results.

2. There are some grammar issues. Please double-check this paper and improve English expression.

3.  The conclusion should only conclude the important results of the work, it is suggested to reduce the content decisively. It is suggested to conduct a combination of quantitative and qualitative analysis.

4. A detailed discussion and link on experimental values is recommended. Authors are recommended to introduce SEM/micrographs of workpiece.  A nice quantitative analysis is there. but a small focus is on experimental data. Please share a balance. 

5. In introduction, a table is recommended where a descriptive analysis of the topic should be included. A comparison of what other author did and how you are doing is welcome.

Good luck.

Author Response

Changes list and responses

Manuscript ID: materials-1855505

Title: Analytical and Numerical Modeling of the Pullout Behavior Between High-strength Stainless Steel Wire Mesh and ECC

Authors: Xu-Yan Zou, Ya-Wen Liu, Jun-Tao Zhu, Ke Li and Jing-Long Cao

The authors sincerely appreciate careful comments of reviewers and editors, which are not only for the improvement of this paper but also bring many insights to the future development of the technique. We have revised the manuscript in accordance with the comments and marked all the amends on our revised manuscript. The following is changes list and responses.

  1. In Abstract, the digital values of thermal measurement should be presented, absorbing the attention of readers. At present, the Abstract is declarative sentences, while the outcomes obtained in this work are lost, especially for the digital results.

Thanks a lot for the reviewer's important suggestion. We have revised the abstract to be more precise. The purpose of this study is to propose a more accurate bond-slip model between high-strength stainless steel wire mesh and ECC, which can be suitably used for numerical simulation. Therefore, a quantitative description of the accuracy of the numerical simulation is added to the abstract, seen the abstract in revised paper.

  1. There are some grammar issues. Please double-check this paper and improve English expression.

Thanks the reviewer very much for these careful works. The manuscript has been thoroughly revised and edited for several times, so we hope these modifications can meet the writing requirements of the paper. If there are still problems, we sincerely hope that the reviewer can give us further suggestions and we will make better modifications.

  1. The conclusion should only conclude the important results of the work, it is suggested to reduce the content decisively. It is suggested to conduct a combination of quantitative and qualitative analysis.

The authors appreciate the reviewer very much that giving us so important suggestion, which will definitely improve our paper and make a clearer conclusion for our readers. The Conclusion section has been rewritten to give a clear and succinct conclusion in the revised paper.

  1. A detailed discussion and link on experimental values is recommended. Authors are recommended to introduce SEM/micrographs of workpiece. A nice quantitative analysis is there. but a small focus is on experimental data. Please share a balance.

Thanks the reviewer's careful work for giving us the good suggestion. In fact, the author's research group has carried out relevant experimental research, and the detailed discussion of the experimental part can be found in the reference [41]: J. Zhu, K. Zhang, X. Wang, K. Li, X. Zou, H. Feng, Bond-Slip Performance between High-Strength Steel Wire Rope Meshes and Engineered Cementitious Composites [J]. Journal of Materials in Civil Engineering, 2022, 34(5): 04022048.  This paper mainly revised the bond-slip model based on the analysis of test results and discusses the feasibility and accuracy of its numerical analysis. Therefore, in this paper, we no longer discuss the test part in detail, but point out the differences of the previously proposed model in the ascending section through the test results. Then the preliminary model is modified based on the theory of damage mechanics, which considered the effect of the strand diameter on the initial tangential stiffness and the damage inside ECC substrate. In addition, the accuracy of the model is further demonstrated on the basis of finite element numerical simulations.

Admittedly, SEM is a good method for microscopic analysis of materials. Considering that the author only has 10 days to revise, it is difficult to complete this work in this aspect. Please forgive the reviewer for the author's shortcomings in this aspect. However, this good suggestion is of great help to the author to carry out subsequent related research.

  1. In introduction, a table is recommended where a descriptive analysis of the topic should be included. A comparison of what other author did and how you are doing is welcome.

Thanks the reviewer very much for giving us so good suggestion. In order to express the research work of this paper more clearly, the author added a workflow diagram in the introduction, seen in Fig.1 in the revised paper.

Round 2

Reviewer 2 Report

Good job for the authors. Every response and correction is statisfied.

Reviewer 3 Report

Authors have revised the manuscript comprehensively.